# Scoping Review: Intergenerational Resource Transfer and Possible Enabling Factors

**DOI:** 10.3390/ijerph17217868

**Published:** 2020-10-27

**Authors:** Eliza Lai-Yi Wong, Jennifer Mengwei Liao, Christopher Etherton-Beer, Loretta Baldassar, Gary Cheung, Claire Margaret Dale, Elisabeth Flo, Bettina Sandgathe Husebø, Roy Lay-Yee, Adele Millard, Kathy Ann Peri, Praveen Thokala, Chek-hooi Wong, Patsy Yuen-Kwan Chau, Crystal Ying Chan, Roger Yat-Nork Chung, Eng-Kiong Yeoh

**Affiliations:** 1The Jockey Club School of Public Health and Primary Care, Faculty of Medicine, The Chinese University of Hong Kong, Hong Kong, China; jennifer.liao@hotmail.com (J.M.L.); patsychau@cuhk.edu.hk (P.Y.-K.C.); chanyingcrystal@link.cuhk.edu.hk (C.Y.C.); rychung@cuhk.edu.hk (R.Y.-N.C.); yeoh_ek@cuhk.edu.hk (E.-K.Y.); 2Internal Medicine, Faculty of Health and Medical Sciences, The University of Western, Perth, WA 6009, Australia; christopher.etherton-beer@uwa.edu.au; 3School of Social Sciences, Faculty of Arts, Business, Law and Education, The University of Western, Perth, WA 6009, Australia; loretta.baldassar@uwa.edu.au (L.B.); adele.millard@uwa.edu.au (A.M.); 4School of Medicine, Faculty of Medical and Health Sciences, The University of Auckland, Auckland 1010, New Zealand; g.cheung@auckland.ac.nz; 5Retirement Policy and Research Centre, Faculty of Business and Economics, The University of Auckland, Auckland 1010, New Zealand; m.dale@auckland.ac.nz; 6Department of Clinical Psychology, Faculty of Psychology, The University of Bergen, 5020 Bergen, Norway; Elisabeth.Flo@uib.no; 7Department of Global Public Health and Primary Care, The University of Bergen, 5020 Bergen, Norway; Bettina.Husebo@uib.no; 8Centre for Methods & Policy Application in the Social Sciences, Faculty of Arts, The University of Auckland, Auckland 1010, New Zealand; r.layyee@auckland.ac.nz; 9School of Nursing, Faculty of Medical and Health Sciences, The University of Auckland, Auckland 1010, New Zealand; k.peri@auckland.ac.nz; 10Health Economics and Decision Science, The University of Sheffield, Sheffield S10 2TN, UK; p.thokala@sheffield.ac.uk; 11Duke-NUS Medical School, National University of Singapore, Singapore 169857, Singapore; wong.chek.hooi@geri.com.sg

**Keywords:** intergenerational relationships, healthcare policy, social capital

## Abstract

We explore the intergenerational pattern of resource transfer and possible associated factors. A scoping review was conducted of quantitative, peer-reviewed, English-language studies related to intergenerational transfer or interaction. We searched AgeLine, PsycINFO, Social Work Abstracts, and Sociological Abstracts for articles published between Jane 2008 and December 2018. Seventy-five studies from 25 countries met the inclusion criteria. The scoping review categorised resource transfers into three types: financial, instrumental, and emotional support. Using an intergenerational solidarity framework, factors associated with intergenerational transfer were placed in four categories: (1) demographic factors (e.g., age, gender, marital status, education, and ethno-cultural background); (2) needs and opportunities factors, including health, financial resources, and employment status; (3) family structures, namely, family composition, family relationship, and earlier family events; and (4) cultural-contextual structures, including state policies and social norms. Those factors were connected to the direction of resource transfer between generations. Downward transfers from senior to junior generations occur more frequently than upward transfers in many developed countries. Women dominate instrumental transfers, perhaps influenced by traditional gender roles. Overall, the pattern of resource transfer between generations is shown, and the impact of social norms and social policy on intergenerational transfers is highlighted. Policymakers should recognise the complicated interplay of each factor with different cultural contexts. The findings could inform policies that strengthen intergenerational solidarity and support.

## 1. Introduction

Social support systems are subject to the potentially confounding influences of demographic trends, economic conditions, social norms, and public policy development [1,2]. On the one hand, supporting and caring for older people has been a prevalent concern across the world. The world is aging, with approximately 600 million people aged ≥ 60 in 2015, and the number is expected to double and rise to two billion or 21% of the total global population by 2050 [3]. Both developed and developing countries have undergone dramatic demographic shifts, with declining fertility and mortality rates in many countries [4,5]. Especially in developed countries, the increasing number of aging people, coupled with a frequently depressed economic situation, has increased pressure on public health and other welfare systems. On the other hand, increasing female participation in the labour market and divorce rates have made caring for children a concern and challenge for many households [4,6,7]. Similar challenges and problems also arise in those countries, mainly developing countries, where a large proportion of young parents in rural areas leave young children and older parents at home for job opportunities in urban cities [8]. It is important to understand the parent–child relationship and the impact of tenacity of family support on both individual and societal levels by studying intergenerational activity, support, and resources exchange.

Family is a pillar of contemporary welfare [9] and is seen as a critical mechanism against insecurity, risk, and crisis [10]. Family structure varies in different cultural contexts; for example, the practice of three or four generations living under one roof is common in Asia but less common in Western countries. However, family structure in Asian countries has increasingly adopted Western influences over time where the loss of traditional multigenerational family structures, increasing divorce rates and step-family formation, and a weakening sense of filial piety make it less possible for both the older generation and the younger generation in need to obtain support from their family [11,12,13,14,15]. Given various new trends in demographic transformation, socio-cultural changes, and family structures, investigating and understanding the nature, extent, and determinants of intergenerational resource transfer behaviours will have important implications for public policy formulation. Though many studies have explored intergenerational transfers, there is a lack of synthesised reviews to present a comprehensive picture of patterns of transfer between generations around the world.

In our review, intergenerational transfers are any family-based support that flows from older to younger generations or vice versa. We would like to examine what is the extent to which resources transferring across generations and under what contextual factors. The aim of this scoping review was to report the operational concept of intergeneration transfer by (1) identifying factors that contribute to intergenerational transfers behaviours across the world and (2) outlining the patterns of resource transfer between generations. The findings provide a relatively comprehensive picture of transfer behaviour patterns internationally to add knowledge to the existing body of literature. The discussion of these findings provides insights for policy formulation to strengthen intergenerational relationships.

## 2. Methods

We followed Arksey and O’Malley’s methodological framework [16] for conducting a scoping review. The framework includes five stages: identifying the research question; identifying relevant studies; selecting studies for inclusion; charting the data; and collating, summarizing, and reporting the results [16].

### 2.1. Eligibility Criteria of Review

To address the research objectives, eligibility criteria were developed as the researchers’ familiarity with the identified papers increased. To be included, papers needed to focus on patterns of resource transfer, including type of resources, direction of transfer, and factors associated with intergenerational transfer behaviour. Articles were included if they were peer-reviewed journal articles published between January 2008 and December 2018, written in English, and used a quantitative design. Articles were excluded if they focused on beliefs/attitudes about intergenerational transfers, exploration of health impacts of intergenerational transfers on caregivers, and addressed determinants of transfers between family members in different generations per se instead of providing information on type and direction of resource transfer. Considering that demographics are highly related to intergenerational transfer behaviour and could drastically change over decades, findings from the most recent studies would provide the most relevant and important implications for policymakers. Therefore, a proscribed period of 10 years, between 2008 and 2018, was adopted. In addition, in view of the quantification of impact about resources exchange between generations, qualitative and mixed-method studies were excluded. There was no limitation on study location.

### 2.2. Electronic Database Search and Relevance Testing

We searched Pubmed, Medline, AgeLine, PsycINFO, Social Work Abstracts, and Sociological Abstracts between January 2008 and December 2018. We also conducted secondary reference searches of all studies included in the review. The search strategies and list of keywords were reviewed by a panel of researchers. A series of key items were identified through background reading and text mining tools, including ‘intergenerational transfer’, ‘private transfer’, ‘family transfer’, and ‘transfer between parent and child’. Terms that have synonymous meanings to ‘support’ were used, including ‘exchange’, ‘support’, ‘assistance’, and ‘reciprocity’. Overall, the search of the four databases in English between January 2008 and December 2018 yielded 6596 references.

### 2.3. Study Selection

Two researchers (JML, CYC) were involved in the process of study selection. After initial screening of titles, abstracts and citation information, records were screened independently and in duplicate by two reviewers, with differences resolved through consensus. Full text articles were obtained of all selected records. Another two researchers independently assessed all full-text articles for eligibility to determine the final included studies. Differences were resolved through consensus.

Out of the original 6596 references, 561 went into the next stage (full-article review). Based on that review, 75 articles were included for further in-depth analysis (Figure 1).

### 2.4. Data Charting

A data-charting form was designed by two researchers, who discussed which variables, key ideas, or themes to include. One researcher charted the data, and the other verified the data for accuracy. The researchers discussed the results or consulted the third researcher for consensus adjudication where there were inconsistencies or disagreements. Data extracted from each article included the authors, the year of publication, study location, methodology (e.g., study design, sample size, characteristics of target population, measures), focus of the study, theoretical framework underlying the study, and factors influencing intergenerational transfer behaviours as identified by the study.

### 2.5. Data Analysis

We grouped the studies by the types of intergenerational transfer content and summarised characteristics of target population, patterns of intergenerational transfer behaviours, and factors influencing intergenerational transfer behaviours.

## 3. Results

Thirty countries across six regions of the world (North America, Europe, Asia, South America, Oceania, and Africa) were represented in publication from 75 studies. The majority were initiated in North American countries (24/75 independently, 9/75 collaborating with other regions), followed by European countries (28/75 independently, 3/75 collaborating with other regions), Asian countries (11 independently, nine with other regions), a South American country (1/75 independently), an Oceanian country (1/75, independently), and an African country (1/75, with Hong Kong).

Since there were more than one ethnic group in the jurisdictions, study populations from 75 studies were from 45 countries and territories in five major regions of the world: North America, Latin America, Asia, Europe, and Oceania. Sixty-six studies were cross-sectional, and 14 studies were longitudinal. In addition to the wide variety of study designs employed, sample sizes ranged widely from 130 to 28,517 participants. Details are shown in Table 1. All of the study analyses included quantitative data analysis and investigated factors that influenced unidirectional and/or bidirectional transfers between generations or populations. The study generation dyads were adult children–elderly parents, middle-aged parents–adult children, and grandparents–grandchildren (Table 2).

### 3.1. Patterns of Intergenerational Transfer Behaviour

#### 3.1.1. Resource Transfer between Generations

Transfers across generations were categorised into three main dimensions: financial (*n* = 54), instrumental (*n* = 56), and emotional *(n =* 25). Financial transfers were defined as money or material support (e.g., food, gifts, clothing, furniture) [45,59]. Instrumental transfers were physical support with daily activities (e.g., cleaning, shopping, cooking) [23], personal care (e.g., nursing, bathing, dressing) [47], or childcare [39]. Emotional transfers meant providing companionship or advice [35], listening [50], or self-reported feeling closeness toward each other [60].

#### 3.1.2. Direction of Intergenerational Transfers

Transfers occurred from one generation to another generation (unidirectionally) or to each other between different generations (bidirectionally). Three types of transfer were examined: upward transfers from younger generation to older generation in their family, downward transfers from older generation to younger generation, and concurrent (bidirectional) transfers between two generations. Studies *(n =* 11) identified different patterns of net flow of intergenerational transfers. In some, it was found that downward flows occurred more frequently than upward flows [21,41,46,77], which means that transfers were more likely to flow from the older generation to the younger generation. In contrast, in other studies, it was reported that transfers from the younger generation to the older generation were more common than the reverse [1,5,14,28,62]. Some studies suggested that when the middle generation was providing support, it tended to be to children over parents, in terms of likelihood and amount [31,32].

#### 3.1.3. Geographical Patterns

There were distinctive geo-cultural differences in intergenerational transfers. Asians received the highest proportion of financial support, while people from North America received the lowest [47]. Studies with European populations showed a difference in transfers between Northern and Southern European countries [6,19,38,39,43,63]. Compared to northern regions, southern regions have lower proportions of grandparents caring for grandchildren [6,39], fewer support relations [19], less support from children to parents [43], and higher likelihood of support from parents to children [63]. However, parents in Southern European countries had higher likelihood of receiving care from their children than their counterparts in Northern European countries [38].

### 3.2. Factors Influencing Intergenerational Transfer Patterns

One study explored possible factors influencing intergenerational transfer behaviour and derived intergenerational solidarity models [6], in which factors were categorised into four main aspects: needs and opportunities, demographic characteristics, family structure, and cultural context. The aspect of ‘need’ indicated need for the intergenerational transfer, for example, a difficult financial situation increased need for support, whereas the ‘opportunity’ aspect indicated predisposing factors (e.g., health status or employment) that reflected opportunities or resources for intergenerational transfer. Demographic factors included gender, age, education, and marital status. Family structure was represented by family composition, family relationship, and living arrangement. Under cultural-political context, ethnic background and social conditions within which intergenerational relationships were developed were included, such as state policies, social norms, family customs, and culture. The summary of intergenerational resource transfers unidirectionally upward to the older generation, unidirectionally downward to the younger generation, or bi-directionally exchange are shown in Figure 2a–c, respectively.

### 3.3. Factors Influencing Upward Transfer Behaviour (Resources to Older Generation)

#### 3.3.1. Needs and Opportunities

A number of studies explored how the older generation’s needs and resources promoted or hindered the occurrence of support from younger generations. These studies investigated older people’s health status, employment status, and financial resources and the availability of other resources to them. Those who reported poor health status or had more health problems (e.g., activities of daily living problems) were more likely to receive different kinds of transfers from their children [1,2,5,10,11,13,18,19,22,23,27,28,31,32,33,37,38,43,45,46,47,50,51,56,59,61,62,78].

People who were employed had less chance of receiving support from their adult children [13,15,62]; unemployment increased elderly parents’ probability of receiving financial support [47,78] but led them to receive less care from children [47]. The older generation’s financial resources had significant influence on transfers to them. Their higher income reduced the probability or the amount of financial transfers [11,24,26,37,46,56,62,75] and emotional support [45]. Besides, the older generation’s assets and wealth decreased the amount of financial transfers from children [11,24]. In addition, the presence of home care or residential care reduced the extent of care provided to parents [26,50].

Adult children with better health status were more likely to provide instrumental transfers [19,28,43]. Children’s employment status had substantial impact on transfers to parents. Children who were employed reported more financial transfers [23,51,59] but less instrumental transfers [19,23,27,38,51] to their parents.

#### 3.3.2. Demographic Factors

Many studies investigated the relationship between the older generation’s demographic factors (e.g., age, gender, marital status, education) and upward transfers from the younger generation. Most studies suggest that transfer receivers’ age was positively related to financial transfers [1,2,11,15,51,67,78], instrumental transfers [1,2,5,15,19,23,28,35,37,43,51], emotional transfers [2,67,68], and overall transfers [27,31,48,58,62,75]. However, Albuquerque (2014) [10] found little impact of age on the probability of older people’s receiving money. It was found that, compared to those aged 60–64, people aged 70 or over were less likely to receive money from their adult children [24]. It was reported that father’s age had a negative impact on receiving instrumental transfers [67].

Studies showed that women tend to receive more transfers of different kinds from their children than men [2,8,11,15,18,21,24,27,31,32,47,48,51,56,62,67,68,71,75,77].

Regarding marital status, widowhood increased the amount of and likelihood of receiving financial, instrumental, and emotional transfers [5,11,18,19,37,40,47,56,59,70,78]. However, the existence of a lived partner was associated with receiving smaller transfers or being less likely to receive transfers [12,24,32,35,37,46,51,60,61,68,75]. In addition, Song et al. (2012) [67] suggested that married people had a greater probability of receiving emotional transfers from their children, particularly daughters. The findings regarding educational factors were inconsistent while several studies showed that the older generation’s education level was positively associated with receiving instrumental, emotional, or financial support [15,21,23,24,26,35,45,47,75,77]; almost an equal number of studies showed opposite results [1,11,14,18,22,23,51].

Studies also highlighted the characteristics of the younger generation transferring resources to the older generation. A majority of studies showed that unemployed children provided less financial [1,72] and emotional support [4] to parents. There was high consistency in findings on the influence of children’s financial resources on transfers to parents. Children’s income or financial resources were positively associated with financial transfers [5,13,25,45,51,58,61,72], instrumental transfers [19,32,45], emotional transfers [45], and overall support [32]. Brandt (2013) [19] revealed that the higher the children’s income, the more likely children provided help of all types to parents. Compared to male adult children, female adult children tended to provide more instrumental support [13,15,18,23,38,40,42,51,55,59], emotional support [4,5,21,59,68], and overall support [22] to parents. The studies examining the relationship of children’s gender and transfers to the parent generation yielded relatively consistent results. However, when it came to financial support, more discrepancy was observed. According to Mureşan (2017) [59], Hu (2017) [42], Ghazi-Tabatabaei and Karimi (2011) [2], and Quashie (2015) [1], daughters provided more money and material to parents than sons did, but Yount et al. (2013) [77], Chen and Jordan (2018) [23], Lin and Yi (2013) [5], Kim et al. (2012) [51], and Wu and Li (2014) [75] disagreed and suggested that sons provided more money to parents. Cheng et al. (2013) [22] found a positive relationship between marriage and overall support to parents. High inconsistency was observed in studies investigating the relationship between marital status and transfers to parents. Five studies [8,23,26,67,77] indicated that married children had a higher tendency to give financial support to parents, and another two studies [23,67] found that marriage increased the likelihood of providing instrumental support, while two studies [51,55] disagreed. One study [68] showed a positive association between marriage and emotional support to parents, while another study [26] showed the opposite association.

The findings related to children’s education were highly consistent. The majority of studies found that the higher the level of children’s education, the more often they made financial transfers [5,8,23,25,26,28,59,64], instrumental transfers [5,19,28,31,40,43,53], and emotional transfers [21,31,35,53,59,67] to parents.

#### 3.3.3. Family Structure

Studies examined the impact of living arrangements (e.g., co-residence), family size, number of children/grandchildren, relationship quality, and relationship with the younger generation. Living arrangements were closely related to intergenerational transfer behaviours. Adult parents who lived as couples received less money [59] and less care [38]; those who lived alone received more time and money [10,27]. Distinctively, instrumental transfers were more frequent in upward flows when the older generation were living with younger generations. Living with the younger generation increased finance and care transfers [15,23,24,47,48,62,67], while living with respondents’ siblings reduced financial transfers [13]. Effects of co-residence were not consistent. Family size appeared to be associated with upward flow of transfers. Elderly parents whose family had two or more generations had a higher probability of receiving money, materials or services from younger generations [14]. Findings related to the number of children were relatively consistent; the number of children was positively associated with monetary support [11,14,24,37,56,75,78], instrumental support [14,51,56], and emotional support [37]. Having one’s own children had a negative influence on upward transfers to parents in the form of money [21,51], instrumental transfers [18,30], and emotional transfers [33]. Having grandchildren had a positive relationship with receiving financial support [78], non-financial support [46], instrumental support [10], and emotional support [15]. When parents and children were non-genetically related, upward transfer occurred less frequently [19]; biological children provided more emotional transfers to parents [68]. In regard to the form of relationship, mother–daughter dyads had positive relationships, with care and practical help to mothers [19,38,43]. More upward transfers occurred when different generations shared similar values [35], emotional closeness [36], and good quality relationships [22,27,31,53].

Geographic distance took a significant toll on upward transfers. Distance between parents and children had a negative impact on instrumental transfer [5,10,13,19,27,31,35,38,40,59], and emotional transfer [12,31,35,68]; however, geographic distance increased financial support from the children’s generation [1,2,11,50,59,67].

#### 3.3.4. Cultural-Political Context

State policy and the welfare system had bearing over transfers across generations. When elder parents were receiving comprehensive social security assistance (CSSA), they received less money from their children [24]. Brand and Deindl (2013) [20] and Brandt (2013) [19] found that the more social services available, the more likely children provided help to parents, but the more social expenditure, the less money children provided money to parents. Igel et al. (2009) [43] made a similar finding: social services increased children’s provision of help or care to parents. Studies showed that where the government provided more social care for older adults, the younger generation provided lower levels of care to their parents [38,59]. Children with a higher sense of obligation or stronger filial norms tended to give more financial transfers as well as non-financial transfers to parents [19,32,59]. Children who saw support as a family responsibility provided more help to parents [43]. Those who viewed religious ceremonies as important provided more emotional transfers to parents [4]. Lin and Yi (2011) [12] compared children’s provision for their parents between China and Taiwan and found that children in China provided more money to their parents as they became older, but the trend was reversed in Taiwan. Non-white mothers received less emotional support from their children [35]; black parents had more support from children [51,68]. Thus, results with regard to race were conflicting.

### 3.4. Factors Influencing Downward Transfer Behaviour (Resources to Younger Generation)

#### 3.4.1. Needs and Opportunities

Children with health problems or disabilities tended to receive more emotional, financial, and instrumental support from their older generation [31,35,59]. The more life problems younger generations encountered, the more support they received from older generations [31,33]. The findings in this category were highly consistent. A number of studies reported a correlation between possessing good health in the older generation and increasing downward financial transfers [6,19,44,57] as well as instrumental transfers, including practical help and caring for grandchildren [18,43,52]. Possessing poor health or health limitations in the older generation had negative impact on parents’ and grandparents’ financial transfers [9,19,20,57,78] and on instrumental transfers, including help, care for grandchildren [9,20,23,28,39,51,52,73], and physical transfers [47]. It was found that older Israeli parents with depressive symptoms or IADL limitations were less likely to be net givers of money and instrumental support. Mudrazija (2014) [60] made a similar finding: parents with health limitation gave less financial support for their children.

Children who were still in education tended to receive more parental support [31,32,33,63]. Children who were employed tended to receive less money [23]. The more income children possessed, the less financial transfers they received from parents [41,53]. Unemployed children received more money, materials, and practical support [1,29,44]. Most studies found that the older generation’s employment was positively associated with downward financial support [21,29,51,63], emotional support [74], and overall support [14] but was negatively related to downward instrumental transfer [1], including offering childcare or grandchild care [73]. Knodel and Nguyen (2015) [52] found that grandparents who worked provided more care to grandchildren in Myanmar but not in Vietnam. Meanwhile, unemployment decreased the amount of financial support [47,78], physical care [47], and emotional support [4] to the younger generation. Older generations’ financial resources had a profound impact on transfers to younger generations. When parents or grandparents had higher income, could make ends meet, and had more financial resources, they were more likely to make financial transfers [1,20,23,29,44,51,63,66] and instrumental transfers like help and care for grandchildren [1,6,19]. It was found that parents’ wealth increased the probability of becoming a net giver of money and instrumental support in Germany but decreased the odds of becoming a net giver of money and instrumental support in Israel [9]. Mudrazija (2014) [60] found that, if parents made ends meet easily, the probability of net value flowing in favour of the children was greater.

Jiménez-Martín and Vilaplana Prieto (2015) [44] found receiving formal care in the older generation, particularly from a private provider reduced the probability of financial provision to adult children. Mudrazija (2014) [60] found that when parents received professional homecare, there was less net value in favour of children in intergenerational transfers.

#### 3.4.2. Demographic Factors

Studies investigated younger generations’ characteristics in upward transfers, including age, gender, marital status, and education level. A significant number of studies reported a positive relationship between children’s age and children’s financial provision [2,5,13,15,26,49,72], instrumental provision [1,2,33,40,45,49,59], and emotional provision [5,49], although few studies found that increasing age decreased financial provision [25,26], instrumental provision [47,55], and emotional provision [4,26,67].

With growing age, children received declining transfers from parents [19,27,29,31,32,33,36,39,42,61,62]. The age of the youngest child in a family was negatively correlated with level of support from parents [49]. Increasing age of grandchildren was linked to declining transfers from the older generation [6,19,73]. Increasing age of the older generation negatively impacted downward financial transfers [9,21,23,47], instrumental help transfers (e.g., childcare for grandchildren) [1,6,9,18,20,28,39,47,52,54,73], emotional transfers [35,68], and overall support [60]. A number of studies showed that female members of the older generation were more involved than male members in downward instrumental transfers [1,18,21,51,52,74], financial transfers [66], emotional transfers [4,21], and overall transfers [60]. It has been reported that men tended to give less downward transfers [32,49]. In addition, granddaughters received more emotional support from grandparents than grandsons did [65]. It was found that women in France had more support from mothers while women in Bulgaria received less help from their mothers [40].

Marital status appeared to be associated with downward transfer behaviour. Two studies found that married children received less emotional support [68] and money [20]. One study found that married children received less financial support but more instrumental support from parents [23]. Children with a single parent in a status of widowhood, divorce, or separation tended to receive more financial and physical transfers from parents [29,47]. Single or unmarried children received more advice, practical support, or financial support from parents [1,21,31,33]. Married parents were more likely to give emotional transfers, financial transfers [47], and instrumental transfers [23,52]. Immigrant women who were divorced or never married provided less childcare to their children than those who were married [73]. Widowhood was negatively associated with financial transfers [60,78], and instrumental transfers including babysitting of grandchildren [1,21,40,60,74,78]. It was found that married middle-aged parents were involved less in downward transfers to their adult children [49].

There was high consistency in studies investigating the relationship between education level and different forms of downward transfer. Higher education level in the younger generation had a positive influence on various forms of downward support, including emotional support [35], instrumental support [28], money [27,28], childcare help [39], and financial help [27]. The higher the level of parents’ or grandparents’ education, the more likely they were to give financial transfers [1,10,20,21,23,29,31,47,57,60,63,66,74,78], instrumental transfers [6,20,23,31,43,48,60], and emotional transfers [21,31,35,68,74]. Only two studies indicated that higher education level decreased the rate of grandparenting engagement in primary childcare activities [18,43].

### 3.5. Family Structures

Studies revealed a consistent relationship between the number of children and downward transfers. The number of children was negatively correlated with downward transfers of emotion [68], money [15,20,28,29], time support [60], likelihood of instrumental help [19,20,28], and grandparenting [18,43]. One-child-family parents gave more financial and instrumental support to their offspring [23]. Having offspring increased the likelihood of children receiving practical support [21,36], advice [31], and money [29,63] from parents. The presence of siblings was negatively related to financial transfers [19,21], instrumental transfers [19], and emotional transfers [59] to younger generations. Only one study [1] indicated a positive relationship between having siblings and instrumental transfers; otherwise, the more siblings, the less financial transfer [25,28,61,72], instrumental transfer [5,19,36,40,43], and emotional transfer [4] given to parents. Parents who lived together provided more support for transportation to their children [36]. Stepchildren were less likely to receive help or money from older generations [19,63]. In families that had a larger number of generations alive, older generations tended to give more emotional support to younger generations [4]. Help and money were more likely to flow from mother to daughters than to sons [19,20].

Regarding living arrangements, co-residence with children or family members was positively associated with various downward transfers, including help and care [47], instrumental support [18,35,53], financial support [23], emotional support [53], grandchild care [52], and overall support [14]. Adult children who lived without partners had more childcare from parents [18,39]. A positive relationship between older and younger generations promoted downward transfers (e.g., instrumental support, advice, finance) [20,31,33,35,36,51,60]. Adult children who shared similar values with their mother received more instrumental support [35].

In addition, compared to son–mother dyads and son–father dyads, upward transfers were found to flow more frequently between daughter and mother [19]. Kim et al. (2012) [51] discovered that children who had higher levels of positive social support were more likely to give financial transfers as well as time transfers to their parents. Evandrou et al. (2018) [30] found that the probability of instrumental transfers to parents was higher when sons and daughters were worried about their parents.

Distance in location of residence between older parents and their adult children negatively impacted downward help, including grandchild care, in terms of probability and intensity [6,18,20,39], along with downward emotional support [35] and instrumental support [10,35]. The greater the geographic distance between middle-aged parents and their children, the less help was provided to children [43]. Co-residence had a positive impact on financial transfers [26,72], instrumental transfers [31], emotional transfers [21,26], and overall transfers [19].

#### Cultural-Political Context

In terms of ethno-cultural factors, a study that explored factors linked to intergenerational social support across the world found that older Latin American and Asian adults had lower propensities to give support to younger generations compared to their North American counterparts [47]. In a study, where downward transfers of African Americans, Hispanics, and non-minority Americans (Floridians) were examined, it was found that African Americans and Hispanics gave less money and gave to fewer children than Americans (non-minorities) [66]. It was found that Hispanics were more likely to give financial transfers to children than African Americans [51]. Bordone and de Valk (2016) [18] found that parents who were of north-western or southern European origin had a higher level of downward support compared to those of all origins. In Israel, the higher the frequency of contact, the higher the probability of being a net giver of money and instrumental support to children [9].

In countries with generous welfare systems, there was a higher probability of downward financial and help transfers [20,28,60]. Belief in parental obligations was positively associated with downward transfers to children [1,49]. Viewing offspring ties as important also increased emotional support to younger generations [33]. A stronger sense of obligation to help additionally predicted more support to younger generations [31].

### 3.6. Factors Influencing Bidirectional Transfer Behaviour (Exchange Resources between Older and Younger Generations)

Few studies discussed concurrent transfer flow between older and younger generations. Receiving financial support [5,13,26,28], instrumental support [5,13,26,30,45,67], or other forms of support [19,22,26,59] from parents had a strong association with children’s transfers to parents. Children who provided support to parents received more financial and material support from parents [1]. Parents who received money or help from children tended to reward children with more help [20]. Parents who received more support from children were more likely to provide babysitting, chores, and financial support to children [74]. Elderly parents who were caregiving for grandchildren [26], providing financial support [1,8,19,23,38,43] or instrumental help [8,23] to children had a higher level and likelihood of transfers from children. Leopold and Raab (2013) [55] revealed that parental transfer to children in long run increased the likelihood of help and care from children.

A higher educational level predicted higher participation in transfers between generations [27,41,53]. One study explored factors that contributed to giving more or receiving more in intergenerational relationships [17]. It found that people aged 25–34 and aged 65–79 were more likely to give finance and practical help than those aged 35–54. People aged 18–24 and aged 25–34 were more likely to receive financial and practical support; males had higher likelihood of giving financial support but gave less practical support. People with higher income tended to give more and receive less transfers from other generations. People who were separated or had sparse contact with other generations tended to not only give more financial transfers but also receive more financial transfers from the other generations. Single parents with children under 18 years old were more likely to give and receive more financial transfers.

## 4. Discussion

By examining studies on intergenerational transfers across the world, a series of factors have been identified under an intergenerational solidarity framework. Demographic factors that contribute to intergenerational transfers included age, gender, marital status, and educational level. Needs and opportunities of both transfer receivers and transfer providers played important roles. Greater need in the form of worse health status, more common unemployment status, lower income, or lower wealth increased receipt of support and decreased giving it. Better health, employment, and higher incomes were opportunities to increase provision of transfers. Family structures (i.e., number of children) exerted a great impact on intergenerational transfers. In countries with an emphasis on filial and family responsibility, intergenerational transfers occurred more frequently; in contrast, in countries with well-developed social welfare systems, the likelihood of intergenerational transfers was lower.

Gender was important in both upward and downward transfers. Women of both older and younger generations provided more instrumental support to the other generation than did men. The traditional gender roles that see housekeeping and childcare as the primary functions of women may have contributed to this phenomenon. Under this circumstance, even as female participation in the workplace has increased, women are expected to undertake large proportions of house chores, care for family members, or other practical help tasks. In particular, in some countries, maintaining a patriarchal gendered division of labour involving more domestic work has exerted pressure on women [79,80].

Our findings are consistent with those on the prevalence of female caregivers in informal systems. Long-term caregiving was most often provided by married women; in addition, 60% of this group were employed and 45% provided monetary support to their parents and in-laws [80,81]. Since the majority of caregiving responsibility falls on women, their wellbeing should be given attention. A scoping review, mapping 55 studies focusing on factors affecting adult children caregivers’ well-being, suggested that daughters were more likely to be subject to negative impacts when providing transfers to their parents [82]. Daughters reported more depression symptoms and suffered higher emotional costs compared with sons [83,84,85]. In addition, grandmothers who gave grandchildren care experienced more mental health problems (i.e., stress, depression symptoms), physical problems, and family-function-related problems [81,86]. The burden and loss encountered by women involved in intergenerational transfers should not be overlooked, since it not only brings them stress and pain but also results in costs for the community by increasing the need for mental and physical healthcare [81]. Hogan (1990) [81] pointed out that resources given by current policy to community elders are insufficient and place burdens on families. Hogan (1990) [81] also suggested that social policy made for caregivers should take emotional strain into consideration and create conditions for them to better take care of family members, including flexibility in the workplace, compensatory time off, information on community resources, and the availability of dependable, professional, supervised social, and in-home services. More even distribution of respite and day-care services and convenient transportation systems were also recommended.

The complicated interplay of ethno-cultural background, culture norms, social welfare systems, and economic situation must be recognised. The role of family members in intergenerational transfers varies across cultures. People in countries with emphasis on cultural norms of being responsible for family members receive less state support [43]; meanwhile, traditional family structures that foster endorsement and effort to live up to filial norms are challenged by rapid modernization and economic development. In East Asia, where filial piety is a predominant social value, social welfare systems developed under the assumption that family rather than the state was the core unit in the provision of support [67]. The declining co-residence rate between older parents and adult children may undermine support patterns, leading to the older generation receiving less support from the younger generation [87]. Under these circumstances, in societies with under-developed social welfare systems, the elderly may not be able to receive sufficient family support or sufficient public assistance.

In addition, countries characterised by strong filial norms but with less developed economies may have different needs for social policy. Selective migration of young people causes skip-generation households, in which older parents undertake the task of childcare for their children’s offspring [69]. Apart from receiving little public assistance [76], these older parents receive inadequate emotional support from their children because of geographic distance. Therefore, policymakers in countries where social norms frame intergenerational patterns should recognise and address the negative impact of the deterioration of traditional family structures on care for the elderly. To support the elderly and lessen the younger generation’s care burden, more social expenditure or social services should be provided to the elderly, such as formal, professional healthcare or service infrastructures. Policymakers should also take young adults’ needs into consideration. Our findings show young people’s flourishing determines the benefits of older people. A well-rounded welfare system should target both generations instead of focusing on one only.

Our review was a scoping review and as such aimed to be as comprehensive as possible within given parameters. We did not make an assessment and subsequent discussion of the methodological quality of papers included; there is no universally accepted and used standard assessment tool. Also, we excluded qualitative papers, so the review may have missed capturing some factors related to intergenerational transfers. The findings provide helpful precursors to systematic reviews and can be used to confirm the relevant eligibility criteria so as to quantify the impact of factors on intergenerational transfers.

## 5. Conclusions

The review has mapped studies exploring factors contributing to intergenerational transfers. Under the intergenerational solidarity framework, factors fall into four dimensions: (1) demographic factors; (2) needs and opportunities structures; (3) family structures; and (4) cultural-contextual structures. Women’s involvement in instrumental transfers is predominant, which suggests that transfer patterns may occur under the influence of traditional gender roles. In addition, our findings indicate that three directions of intergenerational transfer behaviours: downward from older to younger generation, upward from young to older generation, and reciprocal generation transfer. The downward transfers occur more frequently than upward transfers in many developed countries. The findings of the scoping review of the comprehensive evidence on intergeneration resource transfer provide clarity about the operational concept of intergenerational behaviours in a family unit, increase awareness about intergeneration support in health and social service provision, and inform specific research questions in the next step of systematic reviews.

## 6. Implications

The impact of social norms and social policy on intergenerational transfers is highlighted. Policymakers should recognise the complicated interplay of multiple factors and their various socio-cultural contexts.

## Figures and Tables

**Figure 1 ijerph-17-07868-f001:**
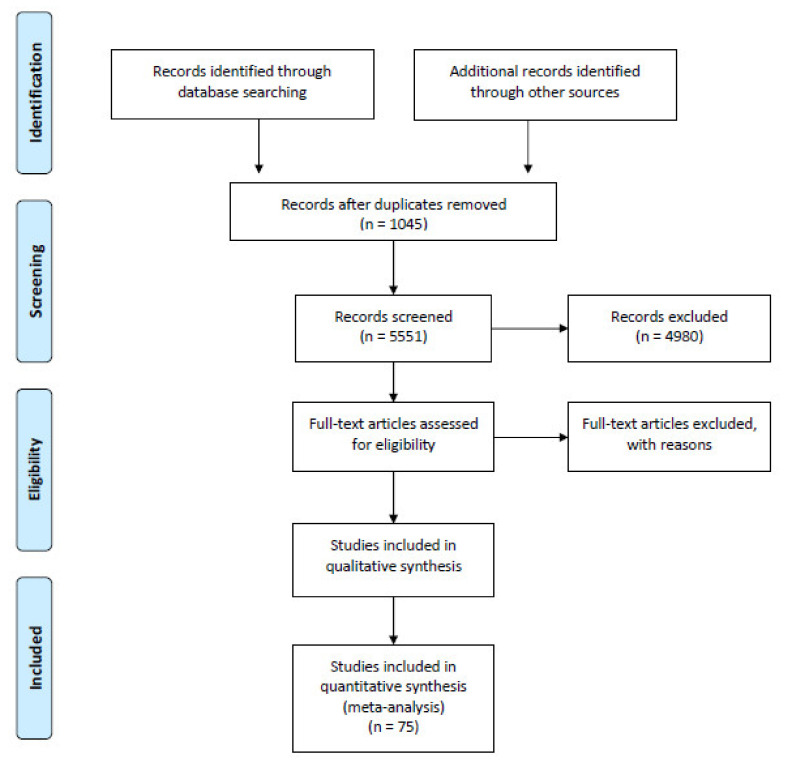
PRISMA 2009 flow diagram.

**Figure 2 ijerph-17-07868-f002:**
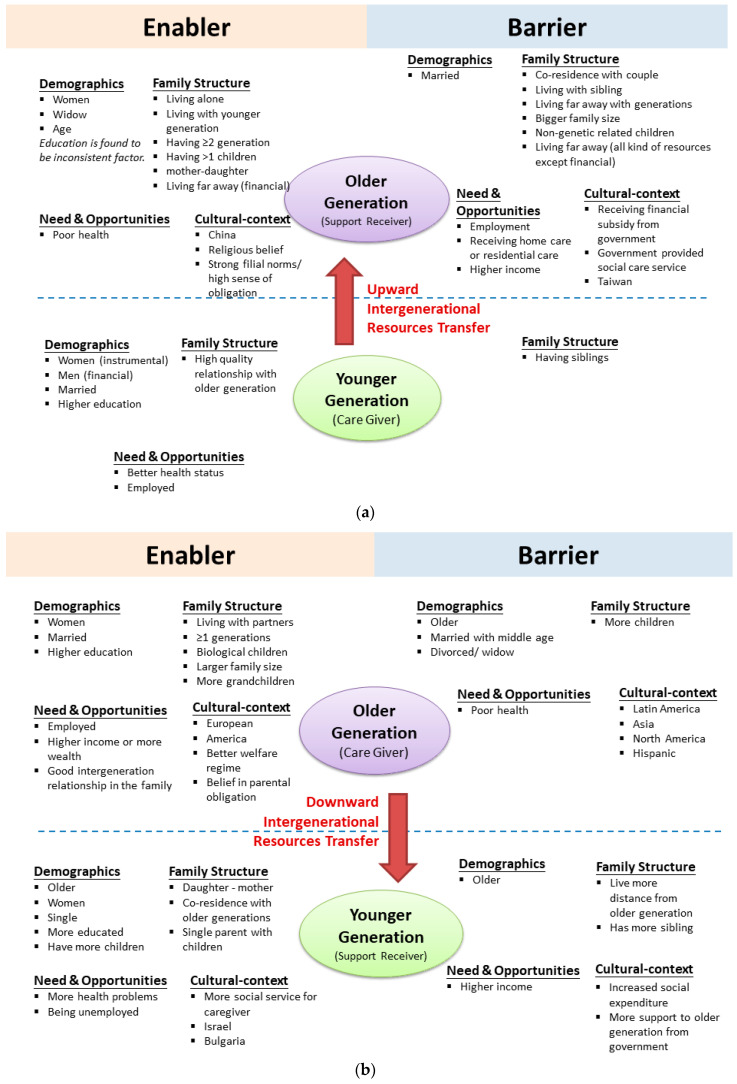
(**a**) Upward intergenerational resources transfer. (**b**) Downward intergenerational resources transfer. (**c**) Bidirectional intergenerational resources transfer.

**Table 1 ijerph-17-07868-t001:** Distribution of included studies (alphabetic order).

Five Regions of the World	45 Countries
Latin America	Brazil, Mexico, Chile, Caribbean
North America	Canada, US
Asia	China, Hong Kong Special Administrative of China, Japan, Singapore, South Korea, Taiwan, Thailand, Egypt, Iran, Saudi Arabia, Vietnam, Israel, India, Indonesia, Malaysia, Philippines, Myanmar, Cambodia
Europe	Austria, Belgium, Bulgaria, the Czech Republic, Demark, France, Germany, Greece, Italy, Ireland, Netherlands, Poland, Sweden, Switzerland, Spain, Slovenia, Russia, Romania, UK, Turkey
Oceania	Australia

**Table 2 ijerph-17-07868-t002:** Information of 75 included studies.

No	Author(s) and Publication Date	Area of the Publication Activity	Origin of Countries	Study Design	Target Population (Sample Size)
1	Albuquerque (2014) [10]	Portugal	Denmark, Sweden, Estonia, Slovenia, Hungary, The Czech Republic, Poland, Germany, Austria, The Netherlands, Belgium, Switzerland, France, Italy, Spain and Portugal.	Cross-sectional	33,647 cases of households with people who have children
2	Björnberg and Ekbrand (2008) [17]	Sweden	Sweden	Cross-sectional	2666 respondents aged 18 years or older
3	Bordone and de Valk (2016) [18]	UK; Austria; Netherlands; Belgium	Denmark and Sweden represent Northern Europe; Austria, Belgium, France, Germany, the Netherlands, and Switzerland represent Central Europe; Greece, Italy, Portugal and Spain represent Southern Europe.	Cross-sectional	62,213 parent-child dyads.
4	Brandt (2013) [19]	Germany	11 European countries (Austria AU, Belgium BE, Denmark, France, Germany, Greece, Italy, the Netherlands, Spain, Sweden, Switzerland)	Cross-sectional	30,000 respondents from 11 European countries;
5	Brandt and Deindl (2013) [20]	Germany	13 European countries (Austria, Belgium, Czech Republic, Denmark, France, Germany, Greece, Italy, the Netherlands, Poland, Spain, Sweden, Switzerland)	Cross-sectional	60,250 dyads
6	Bucx et al. (2012) [21]	Netherlands	Netherlands	Cross-sectional	2022 young adults (ages 18–34 years) in the Netherlands,
7	Cheng et al. (2015) [22]	USA	USA	Cross-sectional	364 parents who had at least one young adult child who also participated in this study
8	Chen and Jordan (2018) [23]	Hong Kong; South Africa	China	Cross-sectional	16,201 parent–child dyads.
9	Chen et al. (2017) [11]	China; UK	China	Cross-sectional	19,947 persons aged 60 and above
10	Chou (2010) [24]	Hong Kong	Hong Kong	Cross-sectional	A total of 4812 household
11	Chou (2008) [25]	Hong Kong	Hong Kong	Cross-sectional	18,877 respondents from 7200 households
12	Cong and Silverstein (2012) [26]	USA	China	longitudinal	1162 parents, aged 60 and older, living in rural areas of Anhui Province, China; 4791 children
13	Cong and Silverstein (2011) [8]	USA	China	longitudinal	Working sample with 1126 elders
14	Cooney and Dykstra (2011) [27]	USA; Netherlands	USA, Netherlands	Cross-sectional	1232 cases for the US sample and 792 cases for the Netherlands sample
15	Deindl and Brandt (2011) [28]	Germany	14 European countries (Austria, Belgium, the Czech Republic, Denmark, France, Germany, Greece, Italy, Ireland, The Netherlands, Poland, Sweden, Switzerland, and Spain)	Cross-sectional	All children aged 50 or more years with at least one natural parent aged 64 or more years alive
16	Emery (2013) [29]	Germany	14 European countries	Cross-sectional	15,412 households from 14 European Countries
17	Evandrou et al. (2018) [30]	UK	UK	Longitudinal	6245 individuals (3073 men and 3172 women)
18	Fingerman et al. (2016) [31]	USA	USA	Longitudinal	191 middle-aged adults (mean age 55.93)
19	Fingerman et al. (2011) [32]	USA	USA	Cross-sectional	The sample included Black (*n* = 216; 108 men and 108 women) and White (*n* = 397; 184 men and 213 women) adults ages 40–60.
20	Fingerman et al. (2011) [33]	USA	USA	Cross-sectional	633 adults aged 40–60 who resided in the Philadelphia Primary Metropolitan Statistical Area
21	Gans and Lowenstein, (2009) [34]	USA; Isreal	Spain, Israel, Germany, England, and Norway	Cross-sectional	6100 respondents
22	Ghazi-Tabatabaei and Karimi (2011) [2]	Iran; Finland	Iran	Cross-sectional	381 adult children
23	Gilligan et al. (2017) [35]	USA	USA	Longitudinal	1338 adult children nested within 369 families.
24	Goodsell et al. (2015) [36]	USA	USA	Cross-sectional	2004 Middletown Kinship Survey (*n* = 338)
25	Guo et al. (2009) [37]	USA	China	Longitudinal	1715 older adults aged 60 and older from the Chaohu region
26	Haberkern and Szydlik (2010) [38]	Switzerland	Austria, Belgium, Denmark, France, Germany, Greece, Italy, The Netherlands, Spain, Sweden, and Switzerland	Cross-sectional	28,516 people
27	Hank and Buber (2009) [39]	Germany	Austria, Denmark, France, Greece, Germany, Italy, the Netherlands, Sweden, Switzerland, and Spain	Cross-sectional	22,000 individuals ages 50 or older
28	Heylen et al. (2012) [40]	Belgium	France and Bulgaria	Cross-sectional	3119 Bulgarian respondents and 2233 French respondents for the analyses on childcare. For personal care, there are 770 Bulgarian respondents and 1557 French respondents
29	Hlebec and Filipovic Hrast (2018) [41]	Slovenia	Slovenia	Cross-sectional	Slovenian national survey of social home care users and their family members, 643 dyads
30	Hu (2017) [42]	China	China	Cross-sectional	A total of 2916 observations (each surveyed older person has multiple children)
31	Igel and Szydlik (2011) [6]	Switzerland	Austria, Belgium, Denmark, France, Germany, Greece, Italy, the Netherlands, Spain, Sweden, and Switzerland	Cross-sectional	28,517 people older than 50 years
32	Igel et al. (2009) [43]	Switzerland	11 countries (Austria, Belgium, Denmark, France, Germany, Greece, Italy, the Netherlands, Spain, Sweden, and Switzerland)	Cross-sectional	The Survey of Health, Ageing and Retirement in Europe, which includes information about 28,517 people
33	Jiménez-Martín and Prieto (2015) [44]	Spain	Austria, Denmark, France, Germany, Italy, the Netherlands, Spain, Sweden, and Switzerland	Longitudinal	13,754 observations
34	Jin et al. (2015) [45]	China; USA	China	Cross-sectional	323 older never-married men; 265 married men under 50, 160 married women, and 518 parents of the above respondents
35	Kalbarczyk-Steclik and Nicinska (2012) [46]	Poland	Austria, Belgium, the Czech Republic, Denmark, France, Germany, Greece, Italy, The Netherlands, Poland, Spain, and Sweden	Cross-sectional	All parents (31,820) whose children are unambiguously identified either as genetically related or non-genetically related (step, foster, or adopted) are selected for the descriptive analysis
36	Khan (2014) [47]	UK	Denmark, France, Germany, Poland, Sweden, UK, Canada, USA, Brazil, Mexico, Russia, Turkey, Saudi Arabia, South Africa, China, Hong Kong, Taiwan, India, Indonesia, Japan, Malaysia, Philippines, Singapore, and South Korea	Cross-sectional	9843 men and 11,390 women
37	Kim et al. (2016) [48]	USA	USA	Cross-sectional	431 middle-aged parents
38	Kim et al. (2015) [49]	USA; Korea	Korea	Cross-sectional	Adults (*n* = 920, age 30–59 years) with at least one living parent and one living parent-in-law
39	Kim et al. (2017) [50]	USA	USA	Longitudinal	380 middle-aged adults
40	Kim et al. (2012) [51]	USA; Korea;	USA	Cross-sectional	20,129 respondents belonging to 6710 respondent households
41	Knodel and Nguyen (2015) [52]	USA; Thailand; Vietnam	Myanmar, Thailand, and Vietnam	Cross-sectional	Sample sizes of person and older were 4080 for the MSOP, 2789 for the VNAS, and 34,173 for the SOPT
42	Komter and Schans (2008) [53]	Netherlands	Netherlands	Cross-sectional	Patterns of reciprocity in intergenerational support exchange among three ethnic groups, ‘Mediterraneans’, ‘Caribbeans’, and native Dutch, (*n* = 3, 520) are analysed.
43	Lee and Bauer (2010) [54]	South Korea	South Korea	Cross-sectional	S nationally representative sample of 3329 grandmothers between 45 and 79 years of age
44	Leopold and Raab (2013) [55]	Germany; Italy	USA	Cross-sectional	1010 respondents comprising 3768 parent–child dyads
45	Lin and Yi (2013) [5]	Taiwan	China, Japan, Korea, and Taiwan	Cross-sectional	1849 valid subjects from China, 1137 from Japan, 1130 from Korea, and 1430 from Taiwan
46	Lin and Yi (2011) [12]	Taiwan	China, Taiwan	Cross-sectional	After deleting subjects with no living aging parent, the final sample (adult children, G2) for China was 1078 respondents, and that for Taiwan was 794 respondents
47	Lin and Pei (2016) [56]	USA; China	China	Cross-sectional	770 elderly parents
48	Li and Shin (2013) [57]	Australia; UK	Urban China	Cross-sectional	903 participants
49	Litwin et al. (2008) [9]	Israel; Germany; Italy;	Germany and Israel	Cross-sectional	The German data: 3020 personal interviews; The Israeli data: interviews 1813 interviews
50	Lorca and Ponce (2015) [58]	Chile	Chile	Cross-sectional	609 people aged 45 and over
51	Moor and Komter (2012) [4]	The Netherlands	Bulgaria, Georgia, Romania, and Russia	Cross-sectional	Bulgaria (*n* = 11,827); Russia (*n* = 10,256); Georgia (*n* = 9858); Romania (*n* = 11,760)
52	Mureşan (2017) [59]	Romania	Bulgaria, Romania, Poland, Czech Republic, Lithuania, Georgia, Russia, Germany, France, and Norway	Cross-sectional	Almost 60,000 cases, of which two third from Eastern Europe (40,512 cases) and one third from Western Europe (19,595 cases).
53	Mudrazija (2014) [60]	USA	Denmark, Sweden, Austria, Belgium France, Germany, the Netherlands, Switzerland, Greece, Italy, and Spain	Cross-sectional	36,095 parent-child dyads from 11 European countries
54	Noel-Miller and Tfaily (2009) [61]	USA; Canada	Mexico	Cross-sectional	1757 couples
55	Quashie and Zimmer (2013) [62]	USA	Latin America, the Caribbean	Cross-sectional	1248 elderly people
56	Quashie (2015) [1]	Thailand	Latin America, the Caribbean	Cross-sectional	1878 households from seven urban cities in Latin America and the Caribbean with persons aged 60 years and over were selected
57	Schenk et al. (2010) [63]	The Netherlands	The Netherlands, Belgium, Austria, Germany, France, Sweden, Denmark, Spain, Italy, and Greece	Cross-sectional	The number of parents in the analysis sample ranged from 947 parents in Denmark to 2006 parents in Belgium, the number of children is 32,758, and they had 17,050 parents in the sample
58	Siennick (2016) [64]	USA	USA	Longitudinal	Data from the National Longitudinal Study of Adolescent Health (N [Wave3] = 14,023; N [Wave4] = 14,361)
59	Sigurđardóttir and Júlíusdóttir (2013) [65]	Iceland	Iceland	Cross-sectional	648 youths; 206 grandparents;
60	Shapiro and Remle (2010) [66]	USA;	USA	longitudinal	6017 households with adult children
61	Song et al. (2012) [67]	China; USA	China	longitudinal	The total number of children–parent pairs was 8064
62	Spitze et al. (2012) [68]	USA	USA	Cross-sectional	Eligible parents were 4215; 7927 observations from the adult children.
63	Strauss (2013) [69]	USA	USA	Cross-sectional	S sample of individuals caring for a parent (n = 77), individuals caring for an in-law (*n* = 26) and a comparison group of non-caregivers (*n* = 1939) from the Midlife Development in the United States study
64	Suitor et al. (2014) [70]	USA	USA	Longitudinal	130 widowed or divorced mothers aged 72–83
65	Svensson-Dianellou et al. (2010) [71]	UK; Greece	Greece	Cross-sectional	190 grandparents around Greece
66	Szinovacz and Davey (2012) [72]	USA	USA	Longitudinal	12,652 respondents; 7702 households
67	Taniguchi et al. (2017) [13]	USA	Japan	Cross-sectional	1158 Japanese respondents
68	Theerawanviwat (2014) [14]	Thailand	Thailand	Cross-sectional	657 elderly persons
69	Vega (2017) [73]	USA	USA	Cross-sectional	29,629 non-Hispanic White, non-Hispanic Asian, and Hispanic women aged 50 years and older
70	Verbrugge and Chan (2008) [74]	USA; Singapore	Singapore	Cross-sectional	1995 national survey (*n* = 4750); 1999 national survey (*n* = 1977)
71	Wu and Li (2014) [75]	China	China	Cross-sectional	1520 observations of residents aged 45 years and above in China
72	Yi and Lin (2009) [76]	Taiwan	Taiwan	Cross-sectional	756 adults
73	Yount et al. (2012) [77]	USA	Egypt	Cross-sectional	886 older adults with living children
74	Zimmer et al. (2008) [15]	USA	Thailand, Cambodia	Cross-sectional	Thailand: 3202 adults aged 60 and older and 17,517 adult children; Cambodian: 777 adults aged 60 and older and 3751 adult children
75	Zuo et al. (2011) [78]	China	China	Longitudinal	895 elder mothers and 759 elder fathers in the working sample

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
