# Peer review of "Scoping Review: Intergenerational Resource Transfer and Possible Enabling Factors"

_ijerph, 2020, doi:10.3390/ijerph17217868_

Round 1
Reviewer 1 Report
The authors of the reviewed paper present a systematic review of n=75 scientific articles dealing with intergenerational resource transfer between the older and the younger generation. They distinguish between three different types of intergenerational resource transfer: upward, downward, and bidirectional transfers. Based on the analyzed studies, the authors identify four categories of factors that promote or hinder intergenerational resource transfer: needs and opportunities, demographic factors like age and gender, family structures, and cultural-contextual structures. The reviewed paper ends with advices for policymakers and outlooks for the future.
I really enjoyed reading this article and thank the authors for the opportunity to review their interesting paper. The topic is timely and relevant to everybody. Time moves on and at some point, you switch from the younger to the older generation. But the intergenerational resource transfer also plays a role beforehand if you have to care for older family members. In summary, the intergenerational resource transfer should play a bigger role in scientific research.
The manuscript is generally well written and structured. However, my biggest overachieving concern relates to scientific soundness of this paper. The authors present several enabling and hindering factors for the intergenerational resource transfer in a very detailed, but only verbal way. They missed to calculate and report statistical parameters like effect sizes based on their study sample (n=75). However, this is important to evaluate the different enabling and hindering factors. For example, the reader may ask him- or herself, whether there are any differences between the factors (e.g., family structure is more important for intergenerational resource transfer than cultural context). Hence, a major weakness of the paper is the lack of statistics. For readers’ convenience the authors should provide at least some statistics that allow a more sophisticated comparison between enablers and barriers. Thus, I urge the authors to calculate effect sizes (or other common statistics) which allow an assessment of the influencing factors on intergenerational resource transfer. These parameters could be presented in a separate table. I do not think this would create an undo burden for the authors and is in line with best practice internationally. The readership should be informed if and how much a specific factor influences the intergenerational resource transfer.
Besides this point, I have some minor comments/suggestions that I hope will help the authors to further develop this line of work:
- General comment: Sometimes, the authors forgot to put a comma at the end of a list. For example, not “[…] societal levels by studying intergenerational activity, support and resources exchange” but rather “[…] societal levels by studying intergenerational activity, support, and resources exchange” (see page 2, line 71). Please check the whole manuscript for a consistent writing style. There are several places in the text where this occurs, but I will not list them all here.
- Figure 1 (page 4): In some boxes the text in Figure 1 is cut off. Please check Figure 1 and provide a corrected version.
- Results (page 5, line 148): a space is missing between “from” and “75”; not “[…] in publication from75 studies.” But rather “[…] in publication from 75 studies.”
- Results (page 5, line 148): Two points should be deleted; not “.. The majority […]” but rather “The majority […]”.
- Results (page 11, line 202): a space is missing between “cultural” and “context”; not “culturalcontext” but rather “cultural context”.
- Needs and opportunities (page 14, line 275): a space is missing between “inconsistent:” and “while”; not “[…] were inconsistent:while several studies […]” but rather “[…] were inconsistent: while several studies […]”.
- Conclusion (page 21, line 639): The order of the four categories vary across the manuscript. Here (page 21, line 639): (1) demographic factors, (2) needs and opportunities, (3) family structures, and (4) cultural-contextual structures; compared to the abstract (page 1, line 39): (1) needs and opportunities, (2) demographic factors, (3) family structures, and (4) cultural-contextual structures; see also the result section (headings in bold). Please correct this and keep a consistent order of the four categories.
Reviewer 2 Report
The authors present an interesting study, with an important topic and they describe quite interesting results. It was my pleasure to review your paper, it was really interesting and valuable. I would like to highlight a few points as recommendations and comments.
Introduction
Please, describe the rationale for the review in the context of what is already known. Explain why the review questions/objectives lend themselves to a scoping review approach. Provide an explicit statement of the questions and objectives being addressed with reference to their key elements
Methods
Indicate whether a review protocol exists; state if and where it can be accessed (e.g., a Web address); and if available, provide registration information, including the registration number.
Please, present the full electronic search strategy for at least 1 database, including any limits used, such that it could be repeated.
List and define all variables for which data were sought and any assumptions and simplifications made.
Remove the word meta-analysis from the flow chart
Results
Due to the high number of data included in each sub-section, please include a summary table at the end of each one.
Conclusions
In my opinion the conclusion section lacks some important data.
Misprints
Line 148 from75 studies. ..
Line 155 Sixty-one studies were cross-sectional, and fourteen studies. Indicate the data in numbers
Table 1. 33. longitudinal
Line 202 culturalcontext.
Line 275 inconsistent:while several studies
Round 2
Reviewer 1 Report
I thank the authors for submitting a revised version of their manuscript entitled “Scoping Review: Intergenerational Resource Transfer and Possible Enabling Factors”.
Honestly, I think the authors did a great job! The quality of the revised manuscript increased a lot in comparison to the first version. I really appreciate authors’ efforts and their ideas in the current version. Thank you!
The authors have responded to all my concerns adequately. I have no further comments. Overall, I am happy with the paper and I believe that this article will make a positive contribution to our field.